# REGRESSION CONFORMAL PREDICTION UNDER BIAS

## ABSTRACT

Uncertainty quantification is crucial to account for the imperfect predictions of machine learning algorithms for high-impact applications. Conformal prediction (CP) is a powerful framework for uncertainty quantification that generates calibrated prediction intervals with valid coverage. In this work, we study how CP intervals are affected by *bias* – the systematic deviation of a prediction from ground truth values – a phenomenon prevalent in many real-world applications. We investigate the influence of bias on interval lengths of two different types of adjustments – symmetric adjustments, the conventional method where both sides of the interval are adjusted equally, and asymmetric adjustments, a more flexible method where the interval can be adjusted unequally in positive or negative directions. We present theoretical and empirical analyses characterizing how symmetric and asymmetric adjustments impact the "tightness" of CP intervals for regression tasks. Specifically for absolute residual and quantile-based non-conformity scores, we prove: 1) the upper bound of symmetrically adjusted interval lengths increases by $2|b|$ where $b$ is a globally applied scalar value representing bias, 2) asymmetrically adjusted interval lengths are not affected by bias, and 3) conditions when asymmetrically adjusted interval lengths are guaranteed to be smaller than symmetric ones. Our analyses suggest that even if predictions exhibit significant drift from ground truth values, asymmetrically adjusted intervals are still able to maintain the same tightness and validity of intervals as if the drift had never happened, while symmetric ones significantly inflate the lengths. We demonstrate our theoretical results with two real-world prediction tasks: sparse-view computed tomography (CT) reconstruction and time-series weather forecasting. Our work paves the way for more bias-robust machine learning systems.

## 1 INTRODUCTION

With the growing application of deep learning algorithms to high-impact applications such as healthcare, finance, and climate science, it is equally crucial to develop methods that can robustly quantify their uncertainties. This is particularly important since deep learning algorithms are known to yield confident yet incorrect prediction values (Guo et al., 2017; Wang, 2023; Niculescu-Mizil & Caruana, 2005). Given a prediction by a learning algorithm on a fresh test example, uncertainty quantification methods typically aim to return a *prediction set* with some guarantee that the true value lies within that set. In particular, a prediction set for a regression problem consists of an interval with lower and upper bounds (Lei et al., 2018; Romano et al., 2019).

Conformal Prediction (CP) is a powerful family of uncertainty quantification methods that is *distribution-agnostic*, i.e., makes no assumptions about the underlying data distribution, and generates prediction sets with guarantees of containing ground truth values with some probability (Angelopoulos & Bates, 2021; Fontana et al., 2023; Shafer & Vovk, 2008; Papadopoulos et al., 2002). For example, split CP is based on collecting a separate (from training) *calibration* dataset containing both ground truth and predicted values from the algorithm of interest. Then, given the algorithm's prediction on a test example, split CP computes a *non-conformity* score quantifying how "unusual" new predictions will be with respect to the calibration dataset, and adjusts the prediction via the empirical quantile of the calibration scores to generate a prediction set.

In practice, to minimize prediction uncertainty, we aim to obtain the tightest possible intervals that maintain valid coverage. Interval tightness (or, inversely, its length) depends on various factors:

Figure 1: **Key Intuition.** Conformal prediction interval lengths computed using symmetric adjustments significantly increase with increasing bias, where bias is defined as the systematic deviation of a prediction from ground truth. On the other hand, those computed using asymmetric adjustments are not affected by bias. We aim to understand how bias impacts symmetrically and asymmetrically adjusted prediction interval lengths.

non-conformity scores, data distributions, and underlying models (Kato et al., 2023). There has been much theoretical work analyzing aspects of interval length, including optimal efficiency (Sesia & Romano, 2021; Vovk et al., 2016; Kiyani et al., 2024; Bai et al., 2022), expected set sizes (Dhillon et al., 2024), conditional and marginal set size differences (Xu & Xie, 2023), and differences in set sizes between oracle and estimated prediction intervals (Xu & Xie, 2023; Lei et al., 2018). Existing empirical work includes investigations into lengths under covariate shift (Tibshirani et al., 2019), skewed distributions (Vilfroy et al., 2024), and heteroskedasticity (Lei et al., 2018; Romano et al., 2019).

One important scenario, however, that has not been investigated is how CP fares for a learning algorithm that is *biased*, i.e., produces predictions that systematically deviate from ground truth values. For example, we can define the bias $b$ of an algorithm as the mean difference between the expected values of its predictions $\hat{Y}$ with respect to ground truth $Y$ over the calibration set:

$$b(Y, \hat{Y}) = \frac{1}{n} \sum_{i=1}^{n} (\mathbb{E}[\hat{Y}_i] - Y_i) \tag{1}$$

Bias is a well-known issue plaguing machine learning models due to various factors such as skewed training distributions (Nandy et al., 2022), sensor drift (Jing et al., 2013; Ying et al., 2007; Piazzo et al., 2015), concept drift (Lu et al., 2018; 2014; Bayram et al., 2022), attrition bias (Lewin et al., 2018), and noisy labels (Ding et al., 2022). We find that large bias inflates the conventional symmetrically adjusted interval lengths (Fig. 1-left) because the intervals must be adjusted equally in both positive and negative directions.

In this paper, we argue that the effects of bias on CP interval lengths can be mitigated by computing intervals with asymmetric adjustments (Linusson et al., 2014; Romano et al., 2019) instead of conventional symmetric adjustments. Asymmetric adjustments allow lower and upper endpoints to be adjusted independently to account for directional bias, maintaining the guarantee that the resulting interval contains the ground truth with high probability (Fig. 1-right). While asymmetric adjustments have been theorized to yield longer interval lengths as a consequence of stronger guarantees (Romano et al., 2019), our work observes that with bias, that may not be the case. We expand the theoretical understanding of CP interval lengths for absolute residual ($L_1$) and quantile adjusted (i.e., Conformalized Quantile Regression or CQR (Romano et al., 2019)) non-conformity scores by analyzing their behavior for symmetrically and asymmetrically adjusted interval lengths under prediction bias. Specifically, we prove the following:

1. The upper bound of symmetrically adjusted interval lengths increases by $2|b|$ (Thm. 2),

2. Asymmetrically adjusted interval lengths are not affected by bias (Thm. 3), and

3. Conditions when asymmetrically adjusted interval lengths are guaranteed to be smaller than those of symmetric adjustments (Cor. 3.1).

Our theoretical results are significant for several reasons. First, existing theory shows that while asymmetric adjustments give stronger coverage guarantees, they also result in slightly longer intervals (Romano et al., 2019). However, contrary to this theory, it has also been empirically observed that asymmetric adjustments yield tighter intervals than symmetric ones (Linusson et al., 2014; Wang et al., 2023; Cheung et al., 2024), but the underlying reasons were unclear. Our work provides a theoretical answer to this phenomenon. Second, our theoretical results suggest that asymmetric adjustments are preferable in practice to symmetric ones when systematic biases are expected, e.g., with sensor drift. However, if symmetric adjustments are desired (e.g., when error distributions are assumed to be symmetric), we also offer a method to achieve tighter symmetric intervals during calibration.

We validate that our theoretical analyses align with synthetic and real prediction tasks. Our synthetic experiments validate our theoretical analyses in the ideal setting when $n$ is large for no skew and skewed distributions. Our real tasks validate our theoretical analyses in two settings: when $n$ is extremely low and time series. Our real tasks deal with bias arising in different contexts: (1) when a medical imaging reconstruction algorithm systematically under- or over-estimates volumes of an anatomical region (computed tomography (CT) reconstruction), and (2) temporal "drift" of values over time (weather forecasting). The contributions of this study give machine learning practitioners fundamental and practical insight into using CP under the common scenario of biased predictions.

## 2 BACKGROUND: SPLIT CONFORMAL PREDICTION (CP)

We focus on a "split" CP setup (Papadopoulos et al., 2002; Lei et al., 2018) in this work, but the same theoretical analysis can be applied to other CP forms and extensions Fontana et al. (2023); Barber et al. (2021). In split CP, we assume a calibration dataset $D_C = \{(\hat{Y}_1, Y_1), ..., (\hat{Y}_n, Y_n)\}$ and test point $\hat{Y}_{n+1}$, where $\hat{Y}_i$ and $Y_i$ represent the $i$-th prediction and ground truth values. The calibration data is separate from a training dataset used to train the ML algorithm of interest. The calibration dataset and test point are assumed to be exchangeable. The goal of CP is to construct a prediction interval $C(\hat{Y}_{n+1}) = [L(\hat{Y}_{n+1}), U(\hat{Y}_{n+1})]$ for $\hat{Y}_{n+1}$, where $L(\hat{Y}_{n+1}), U(\hat{Y}_{n+1}) \in \mathbb{R}$ are lower and upper bounds, such that $\mathbb{P}[Y_{n+1} \in C(\hat{Y}_{n+1})] \geq 1 - \alpha$, for some user-specified mis-coverage rate $\alpha \in (0, 1)$. To compute symmetric intervals, we perform the following steps. First, for each data point in the calibration set $D_C$, we compute non-conformity scores $S = \{s_1, ..., s_n\}$. Next, we compute the $(1 - \alpha)$-th empirical quantile of the non-conformity scores $q = Q_{1-\hat{\alpha}}(S)$, where $\hat{\alpha} = \frac{\lfloor \alpha(n+1) \rfloor}{n+1}$ denotes the finite-sample adjusted mis-coverage rate. Finally, we adjust the predictions of the test data using $q$ to achieve valid prediction sets. This algorithm provides marginal coverage: on average, the prediction sets contain ground truth $(1-\alpha)\%$ of the time. More rigorously, based on key CP results:

**Lemma 1** *Let $(\hat{Y}_i, Y_i) \in \mathbb{R} \times \mathbb{R}, i = 1, ..., n + 1$ be exchangeable random variables. Assume that a predictor $f$ has been trained on a proper training set independent of and exchangeable with these $n + 1$ points. Consider a calibration set $(\hat{Y}_i, Y_i)_{i=1}^{n}$ and a fresh test point $\hat{Y}_{n+1}$. Let $s_i$ be a non-conformity score computed using the predictor $f$ for $i = 1, ..., n + 1$. Let $q = Q_{1-\hat{\alpha}}(\{s_i\}_{i=1}^{n})$ be the $\lceil (1 - \alpha)(n + 1) \rceil$-th smallest value of $\{s_i\}_{i=1}^{n}$ and $C(\hat{Y}_{n+1}) = \{y \in \mathbb{R} : s_{n+1} \leq q\}$ be the prediction set for the test point $\hat{Y}_{n+1}$. Then, for any $\alpha \in (0, 1)$:*

1. $\mathbb{P}[Y_{n+1} \in C(\hat{Y}_{n+1})] \geq 1 - \alpha$ *and*

2. $\mathbb{P}[Y_{n+1} \in C(\hat{Y}_{n+1})] \leq 1 - \alpha + \frac{1}{n+1}$ *if random variables $Y_1, ..., Y_{n+1}$ are almost surely distinct*

*Proof: See variations in Vovk et al. (2005); Lei et al. (2018); Tibshirani et al. (2019); Oliveira et al. (2024)*

Many non-conformity scores exist (Kato et al., 2023), including absolute residuals ($L_1$) (Papadopoulos et al., 2002) and quantile-based (Conformalized Quantile Regression or CQR (Romano et al., 2019)) scores.

This can be extend to asymmetric adjustments (Linusson et al., 2014; Cordier et al., 2023) by computing $(1-\alpha_{lo})$-th and $(1-\alpha_{hi})$-th empirical quantiles of the conformity scores, where $\alpha_{lo}$ and $\alpha_{hi}$ are lower and upper mis-coverage rates. In the asymmetric case, the empirical quantiles for the lower and higher mis-coverage rates $\alpha_{lo}$ and $\alpha_{hi}$ are given by $\hat{\alpha}_{lo} = \frac{\lfloor \alpha_{lo}(n+1) \rfloor}{n+1}$ and $\hat{\alpha}_{hi} = \frac{\lfloor \alpha_{hi}(n+1) \rfloor}{n+1}$. It is easy to see that when $\alpha_{lo} + \alpha_{hi} = \alpha$ the asymmetric case yields empirically larger coverage based on the finite sample adjustment. The reason is due to the "rounding effect" of the ceiling function, which tends to push the empirical quantiles toward more extreme values (larger or smaller), especially when $n$ is small. Since the asymmetric case deals with two separate quantiles, this effect is compounded, leading to a prediction set that empirically offers larger coverage. However, as we will see, symmetrically adjusted interval lengths may not always be shorter than asymmetric ones in the presence of bias.

## 3 THEORETICAL ANALYSIS

We assume biased predictions $\hat{Y}_i^b = \hat{Y}_i^0 + b$ where $b \in \mathbb{R}$ is a constant (positive or negative) applied between all unbiased predicted values ($\hat{Y}^0$) and ground truth values ($Y$) from the calibration set. For example, for prediction and ground truth distributions that are symmetric and centered around their means, we can use Eq. 1. However, when the distributions are skewed, the mean may no longer be a good measure of central tendency (Rousseeuw & Hubert, 2011; Huber & Ronchetti, 2011). In Sec. 3.1, we will use results from our theoretical analyses to estimate bias more accurately in these cases.

We consider symmetric non-conformity scores with canonical expression:

$$s_i^b = \max(f_{lo}(\hat{Y}_i^b) - Y_i, Y_i - f_{hi}(\hat{Y}_i^b)), \tag{2}$$

where $f_{lo}$ and $f_{hi}$ are the lower adjustment and upper adjustment functions that have linear properties: $f_{lo}(\hat{Y}_i^b) = f_{lo}(\hat{Y}_i^0) + b$ and $f_{hi}(\hat{Y}_i^b) = f_{hi}(\hat{Y}_i^0) + b$. Eq. 2 covers the conventional $L_1$ and CQR non-conformity scores. For the $L_1$ non-conformity score given by $s_i^b = |Y_i - \hat{Y}_i^b|$, $\hat{Y}_i^b$ represents a point estimate, and the score can be rewritten as $s_i^b = \max(\hat{Y}_i^b - Y_i, Y_i - \hat{Y}_i^b)$. For the CQR non-conformity score given by $s_i^b = \max(Q_{\alpha_{lo}}(\hat{Y}_i^b) - Y_i, Y_i - Q_{1-\alpha_{hi}}(\hat{Y}_i^b))$, $\hat{Y}_i^b$ represents a set of samples $\hat{Y}_i^b = \{\hat{Y}_{ij}^b\}_{j=1}^{n_s}$. The adjustment is given by $q^b = Q_{1-\hat{\alpha}}(\{s_i^b\}_{i=1}^n)$, the prediction interval is given by $C(\hat{Y}_{n+1}^b) = [f_{lo}(\hat{Y}_{n+1}^b) - q^b, f_{hi}(\hat{Y}_{n+1}^b) + q^b]$, and the interval length is given by $L_{sym}(\hat{Y}_{n+1}^b) = f_{hi}(\hat{Y}_{n+1}^b) - f_{lo}(\hat{Y}_{n+1}^b) + 2q^b$. This setup does not cover locally adaptive non-conformity scores (Papadopoulos et al., 2008; 2011; Lei et al., 2018) and variations of CQR such as CQR-r and CQR-m non-conformity scores (Sesia & Candès, 2020). Using Eq. 2, we first derive an upper bound for symmetrically adjusted interval lengths under bias (Thm. 2):

**Theorem 2** *Given biased predictions for a fresh test point $\hat{Y}_{n+1}^b = \hat{Y}_{n+1}^0 + b$, the upper bound on prediction interval lengths of non-conformity scores described in Eq. 2 is:*

$$L_{sym}(\hat{Y}_{n+1}^b) \leq L_{sym}(\hat{Y}_{n+1}^0) + 2|b|, \tag{3}$$

*where $L_{sym}(\hat{Y}_{n+1}^b)$ and $L_{sym}(\hat{Y}_{n+1}^0)$ are the interval lengths computed using symmetric adjustments for predictions with and without bias.*

*Proof: See App. A.1*

We find the upper bounds of symmetrically adjusted interval lengths increase linearly with the magnitude of bias. Next, we show 1) that asymmetric adjustments are not affected by bias and 2) conditions when using asymmetric adjustments produce shorter lengths than symmetric adjustments. To accomplish this, we introduce a similar canonical expression for asymmetric non-conformity scores:

$$(s_{i,lo}^b, s_{i,hi}^b) = (f_{lo}(\hat{Y}_i^b) - Y_i, Y_i - f_{hi}(\hat{Y}_i^b)), \tag{4}$$

---

**Algorithm 1** Estimating bias by minimizing symmetrically-adjusted interval lengths; Example using gradient descent.

---

**Require:** $\gamma$: learning rate, $\{(\hat{Y}_i^b, Y_i)\}_{i=1}^n$: calibration dataset, $\tau$: tolerance, $L_{sym}(\hat{Y}_i^b)$: function that computes symmetrically adjusted interval lengths.

Initialize losses $l_{prev}$ and $l$ s.t. $l_{prev} \geq l$
Initialize $b_{eff}$. E.g., $b_{eff} \leftarrow \frac{1}{n}\sum_{i=1}^n (\mathbb{E}[\hat{Y}_i^b] - Y_i)$
**while** $|l_{prev} - l| \geq \tau$ **do**
    $l_{prev} \leftarrow l$
    $l \leftarrow max(\{L_{sym}(\hat{Y}_i^b - b_{eff})\}_{i=1}^n)$
    $b_{eff} \leftarrow b_{eff} - \gamma\nabla(l)$
**end while**

---

where $s_{i,lo}^b$ and $s_{i,hi}^b$ represent lower and upper non-conformity score adjustments when predictions are biased with $b$. The lower and upper asymmetric adjustments are computed by taking the $(1 - \alpha_{lo})$-th and $(1 - \alpha_{hi})$-th empirical quantile of the sets of non-conformity scores $q_{lo}^b = Q_{1-\hat{\alpha}_{lo}}(\{s_{i,lo}^b\}_{i=1}^n)$ and $q_{hi}^b = Q_{1-\hat{\alpha}_{hi}}(\{s_{i,hi}^b\}_{i=1}^n)$. Thus, the prediction interval is $C(\hat{Y}_{n+1}^b) = [f_{lo}(\hat{Y}_{n+1}^b) - q_{lo}^b, f_{hi}(\hat{Y}_{n+1}^b) + q_{hi}^b]$.

Using this setup, we prove the following relationship for the length of a CP prediction interval using asymmetric non-conformity scores, under bias $b$:

**Theorem 3** *Given biased predictions for a fresh test point $\hat{Y}_{n+1}^b = \hat{Y}_{n+1}^0 + b$, the lengths for $L_1$ and CQR non-conformity scores computed using asymmetric adjustments are bias-independent:*

$$L_{asym}(\hat{Y}_{n+1}^b) = L_{asym}(\hat{Y}_{n+1}^0) \tag{5}$$

*where $L_{asym}(\hat{Y}_{n+1}^b)$ and $L_{asym}(\hat{Y}_{n+1}^0)$ are the interval lengths computed using asymmetric adjustments for predictions with and without bias.*

*Proof: See App. A.2*

We find that asymmetric adjustments are not affected at all by a constant bias $b$, which is a desirable property. However, recall that when predictions are unbiased, asymmetrically adjusted intervals tend to be longer than symmetric ones (Romano et al., 2019). This raises the question: at what level of bias does this behavior reverse? We derive conditions under which, in the presence of bias, asymmetrically adjusted intervals become shorter than symmetric ones:

**Corollary 3.1** *For $L_1$ and CQR non-conformity scores, asymmetric adjustments produce smaller interval lengths than symmetric adjustments under the following condition:*

$$2|b| \geq L_{asym}(\hat{Y}_{n+1}^0) - L_{sym}(\hat{Y}_{n+1}^0). \tag{6}$$

*where $b$ is the bias, $L_{sym}(\hat{Y}_{n+1}^0)$ and $L_{asym}(\hat{Y}_{n+1}^0)$ are lengths computed using symmetric and asymmetric adjustments for predictions without bias.*

*Proof: The result is derived by using Thm. 3, setting $L_{asym}(\hat{Y}_{n+1}^b) \leq L_{sym}(\hat{Y}_{n+1}^b)$, substituting Eq. 3, and rearranging the inequality.*

We find that when the difference in lengths for predictions without bias is greater than 2 times the magnitude of bias, the asymmetrically adjusted interval lengths will be guaranteed to be shorter than symmetric ones.

Our theoretical analyses (Thm. 2, Thm. 3 and Cor. 3.1) provide important insights into how bias affects length and the conditions which asymmetric adjustments yield shorter lengths than and symmetric adjustments. In practice, when $\alpha_{lo} + \alpha_{hi} = \alpha$ and $n$ is large, the interval lengths under no bias are approximately equal $L_{asym}(\hat{Y}_{n+1}^0) \approx L_{sym}(\hat{Y}_{n+1}^0)$, and the lengths for predictions with bias are shorter for asymmetric compared to symmetric adjustments $L_{asym}(\hat{Y}_{n+1}^b) \leq L_{sym}(\hat{Y}_{n+1}^b)$ when $|b| > 0$.

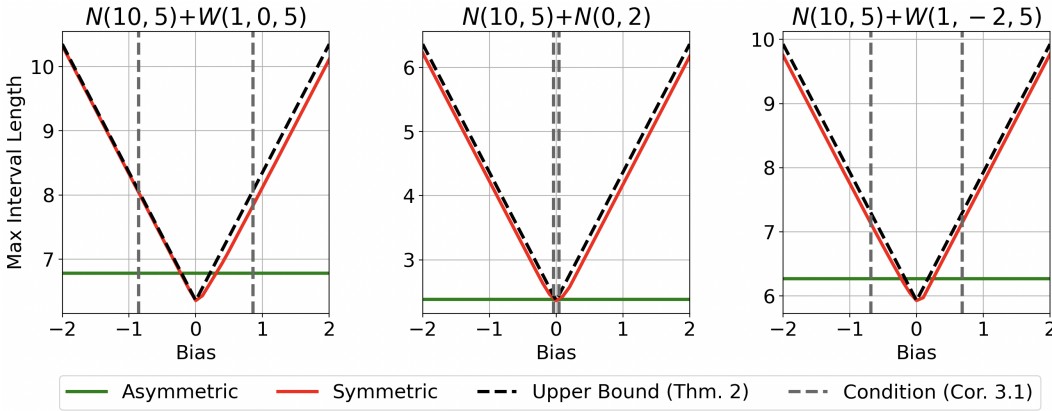

Figure 2: **Synthetic experiments with skewed and noisy predictions align with theoretical analysis.** We set $N(10, 5)$ as the ground truth distribution and added $W(1, 0, 5)$ (right skew), $N(0, 2)$ (no skew), and $-W(1, -2, 5)$ (left skew) to simulate imperfect predictions. The parameter descriptions can be found in the *scipy.stats* documentation. We plot the bias versus the maximum length for symmetric CQR (red) and asymmetric CQR (green). We also plot the theoretical upper bound (Thm. 2, dashed grey) and the value when lengths with asymmetric adjustments are smaller than those with symmetric adjustments (equality in Cor. 3.1, black).

### 3.1 ESTIMATING BIAS

One unanswered question is how to empirically determine bias $b$ given data. To do so, we leverage Thm. 2 and adjust the predictions by a scalar value $b_{eff}$ to minimize the maximum symmetrically adjusted interval lengths:

$$b_{eff} = \arg\min_{C} \left[ \max(\{L_{sym}(\hat{Y}_i^b - C)\}_{i=1}^n) \right] \tag{7}$$

$L_{sym}(\hat{Y}_i^b - b_{eff})$ achieves the minimum length for symmetric adjustments (when $b = 0$). $b_{eff}$ can be thought of as the "debiasing" constant for biased predictions $\hat{Y}^b$. We prove that the objective function in Eq. 7 reduces to minimizing a vertically and horizontally translated absolute value function (App. A.2.1). Therefore, the objective function is convex, and Alg. 1 converges using gradient descent and its variants. For our experiments, we implemented a PyTorch version for CQR-based and $L_1$ scores available at [redacted], and optimize using AutoGrad (Paszke et al., 2017).

## 4 EXPERIMENTS

We next evaluate Thm. 2 and 3 and Cor. 3.1 for $L_1$ and CQR non-conformity scores with synthetic and real-life experiments. For synthetic experiments, we assume normally distributed ground truth data, and simulate predictions by adding different types of noise. For real-life experiments, we consider two scenarios: when data is scarce (CT reconstructions for downstream radiotherapy planning using CQR) and when data is temporally varying (time series weather forecasting using $L_1$). The data distributions can be found in App. B.

### 4.1 SYNTHETIC DATA

We first demonstrate the validity of the theoretical analysis Sec. 3 for CQR using Gaussian $N$ and Weibull $W$ distributions to simulate estimate, ground truth, and noise distributions. We used $N(10, 5)$ to simulate a ground truth distribution. We added noise characterized by $W(1, 0, 5)$, $N(0, 2)$, and $-W(1, -2, 5)$ to the ground truth samples to simulate left-, no-, and right-skewing predictions. We used 1000 calibration data points, 1000 test data points, and 1000 samples per data point to estimate the quantiles. We set $\alpha = 0.1$ for symmetric adjustments, and $\alpha_{lo} = \alpha_{hi} = 0.05$

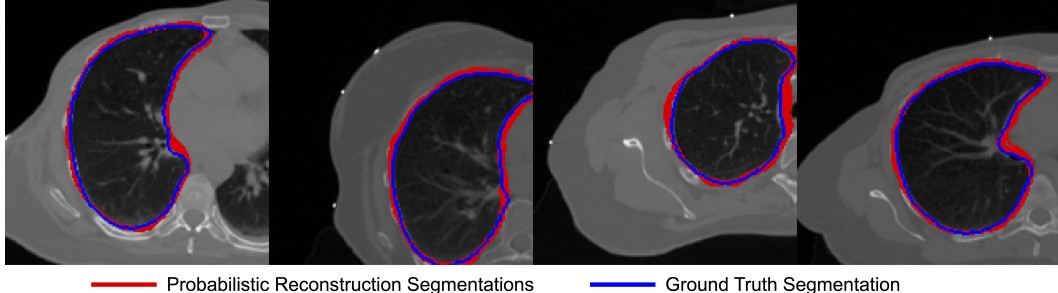

Figure 3: **Biases from using sparse-view CT reconstructions for a downstream segmentation task.** We show slices of 4 different patient ground truth CT volumes. Each slice is overlaid with right lung segmentations from 10 probabilistically sampled reconstructions (red) and segmentations of the ground truth right lung (blue). The reconstructed right lung segmentations consistently overestimate organ volumes compared to the ground truth segmentations. See App. C for experiment details.

for asymmetric adjustments. After determining $b_{eff}$ using Alg. 1, we added a constant bias term from $-2$ to 2 to the debiased predictions to examine the effect of biased predictions on lengths (Fig. 2).

Results in Fig. 2 confirm our theoretical analyses. We find that symmetrically adjusted interval lengths (red) are always upper bounded by the sum of length at $b = 0$ and $2|b|$ (dashed black) holds true (Thm 2). We find that asymmetrically adjusted interval lengths (green) do not change under bias (Thm. 3) and that they are always smaller than symmetrically adjusted lengths (red) when Cor. 3.1 is true (dashed grey).

## 4.2 REAL DATA

Next, we validate our theoretical analyses in two different real data scenarios: where upstream image reconstruction tasks may not fully capture spatial dependencies in downstream metrics (sparse-view computed tomography (CT) reconstruction), and where predictions "drift" from the ground truth over time (time series weather forecasting). Through these two examples, we aim to show the validity and usefulness of our theoretical analysis from different perspectives.

### 4.2.1 LIMITED DATA

In scenarios with limited imaging capabilities, such as low-resource clinics (Aggarwal et al., 2023; Court et al., 2023; Kisling et al., 2018), reconstruction algorithms work with observations that do not contain complete information. For example, sparse cone-beam CT algorithms use limited ($< 100$ instead of the standard 100s) 2D X-ray observations to generate 3D CT scans (Sun et al., 2023; Ying et al., 2019; Shen et al., 2019). The observed information is insufficient to recover the true image with complete certainty, leading to potential biases such as systematically over- or under-estimating organ volumes.

We simulate a medical imaging pipeline, where a patient is imaged using sparse-CT, an image reconstruction algorithm is applied to the projections, and the resulting volume is used for downstream radiotherapy planning (RT). We use Neural Attenuation Fields (NAF) (Zha et al., 2022), a self-supervised image reconstruction algorithm. We synthetically injected noise to the projections, reconstructed the volumes using different initializations of the reconstruction algorithm, and generated plans using the Radiation Planning Assistant (RPA, FDA 510(k) cleared) [1]. More details about our experimental setup can be found in App. C.

To validate our theoretical analysis in Sec. 3 holds true even for extremely low $n$, we use 19 patients for calibration and 1 patient for testing. We generate 10 reconstructions per patient by perturbing acquisition angles, injecting noise into the projections, and using random initializations of NAF. We

---

[1]RPA is a web-based tool that combines organ segmentation algorithms and physics simulations for RT planning.

| Metric | $b_{eff}$ | $\mu(L_{asym}(\hat{Y}^b_{n+1})$ $-L_{sym}(\hat{Y}^b_{n+1}))$ | $P(L_{asym}(\hat{Y}^b_{n+1})$ $\leq L_{sym}(\hat{Y}^b_{n+1}))$ | Cor. 3.1 |
|---|---|---|---|---|
| Heart $D_0$ (Gy) | -0.31 | 334.41 | 0.05 | ✗ |
| Heart Volume ($cm^3$) | -19.74 | 365.40 | 0.05 | ✗ |
| Right Lung $V_{20}$ (%) | -9.47e-3 | 0.02 | 0.05 | ✗ |
| Right Lung $D_{35}$ (Gy) | -1.13 | 1.11 | 0.05 | ✗ |
| Right Lung Volume ($cm^3$) | 66.11 | -108.98 | 1.0 | ✓ |
| Left Lung Volume ($cm^3$) | 58.17 | -92.89 | 1.0 | ✓ |
| Left Lung $D_0$ (Gy) | -0.14 | -0.04 | 0.90 | ✓ |
| Body Volume ($cm^3$) | -287.48 | -564.53 | 1.0 | ✓ |

Table 1: **Real life application of sparse-view computed tomography for downstream radiotherapy planning reveals prevalent biases in predictions and validate bias conditions in Cor. 3.1.** We use a variety of downstream RT planning metrics, including max dose to the heart (Heart $D_0$), heart volume, volume of right lung receiving 20Gy of dose (Right Lung $V_{20}$), dose to 35% relative volume of the right lung (Right Lung $D_{35}$), right lung volume, left lung volume, max dose to left lung (Left Lung $D_0$), and volume of the body. We show the mean difference in asymmetrically and symmetrically adjusted interval lengths $\mu(L_{asym}(\hat{Y}^b_{n+1}) - L_{sym}(\hat{Y}^b_{n+1}))$, effective bias $b_{eff}$, the probability that asymmetrically adjusted interval lengths are greater than that for symmetric adjustments $P(L_{asym}(\hat{Y}^b_{n+1}) \leq L_{sym}(\hat{Y}^b_{n+1}))$, and whether Cor. 3.1 is true (✓) or false (✗).

perform leave-one-out cross-validation to examine each patient's interval lengths when calibrated with the rest of the patients. We use $\alpha = 0.15$ for symmetric adjustments and $\alpha_{lo} = \alpha_{hi} = 0.075$ for asymmetric adjustments, corresponding to $\hat{\alpha} = 0.0567$ for the symmetric case and $\hat{\alpha}_{lo} = \hat{\alpha}_{hi} = 0$ for the asymmetric case. In the asymmetric case, this corresponds to an extreme case of taking the maximum and minimum non-conformity scores.

We show results in Tab. 4.2.1 for a variety of downstream RT planning metrics, including max dose to the heart (Heart $D_0$), heart volume, volume of right lung receiving 20Gy of dose (Right Lung $V_{20}$), dose to 35% relative volume of the right lung (Right Lung $D_{35}$), right lung volume, left lung volume, max dose to left lung (Left Lung $D_0$), and volume of the body. These metrics have important implications for patient safety. For example, in our setup, if a heart $D_0$ is $< 5$Gy or right lung $V_{20}$ is $< 35\%$, the plan is unsafe for the patient. We look at the mean difference between asymmetrically and symmetrically adjusted interval lengths $\mu(L_{asym}(\hat{Y}^b_{n+1}) - L_{sym}(\hat{Y}^b_{n+1}))$, effective bias $b_{eff}$ computed using Alg. 1, the probability that asymmetrically adjusted interval lengths are greater than that of symmetrically adjusted $P(L_{asym}(\hat{Y}^b_{n+1}) \leq L_{sym}(\hat{Y}^b_{n+1}))$, and whether Cor. 3.1 is true ✓ or false ✗(Tab. 4.2.1). Results in Tab. 4.2.1 reveal that for many downstream tasks like segmentation (Fig. 3), predictions could be highly biased (column 2). Moreover, results in Cor. 3.1 can be reliably used to determine whether asymmetrically adjusted interval lengths are shorter than those of symmetric adjustments (columns 4 and 5). When $P(L_{asym}(\hat{Y}^b_{n+1}) \leq L_{sym}(\hat{Y}^b_{n+1}))$ tends to 1, the condition in Cor. 3.1 is met, and vice versa. Our experimental results show that our theoretical analyses are robust to scenarios even with extremely low $n$.

### 4.2.2 TIMES SERIES

Weather forecasting is important for many aspects of daily life, from public safety to agriculture to disaster preparedness and response. We use the Yandex Weather Prediction dataset and the average pre-trained CatBoost model from Angelopoulos & Bates (2021) to predict temperature changes. The temporal dependencies between points violate the exchangeability assumption. Therefore, we use weighted conformal prediction where we use a different adjustment for each new data point (Tibshirani et al., 2019). We use the $L_1$ non-conformity score and weight the data points in the window of size $K = 1000$ equally. This setup effectively reduces to split CP applied each at time window and the theoretical analyses in Sec. 3 apply. The symmetric non-conformity score for time $t$ is given by $s^b_t = |\hat{Y}^b_t - Y^0_t|$. The asymmetric non-conformity scores for time $t$ are given by $s^b_{t,lo} = \hat{Y}^b_t - Y^0_t$ and $s^b_{t,hi} = Y^0_t - \hat{Y}^b_t$. We set $\alpha = 0.1$ and inject an increasing negative bias to the unbiased predicted values $\hat{Y}^b_t = \hat{Y}^0_t - (2 \times 10^{-4})t$. We plot temperature over time for predictions $\hat{Y}^b$ and ground

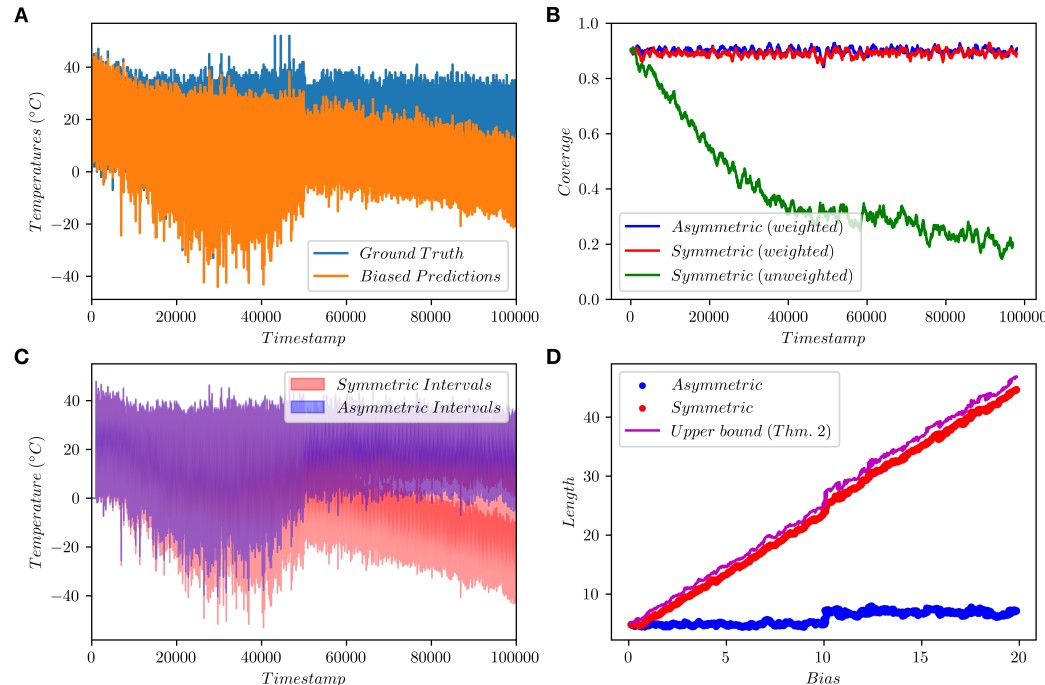

Figure 4: **Real-life application of weather forecasting shows even if predictions drift "far away" from the ground truth values, asymmetrically adjusted intervals are still able to maintain the same tightness and validity of intervals as if the drift had never happened.** We plot A) Temperature over time for biased predictions $\hat{Y}^b$ and ground truth $Y$, B) Coverage over time for weighted conformal prediction with asymmetric adjustments (blue) and symmetric adjustments (red), and naive (unweighted) conformal prediction (green). C) Intervals with symmetric adjustments (red), asymmetric adjustments (blue), and where they overlap (purple), and D) Bias versus lengths for symmetric adjustments (red), lengths for asymmetric adjustments (blue), and upper bound length for symmetric adjustments from Thm. 2 (purple).

truth $Y$ (Fig. 4A), coverage over time for weighted (symmetric and asymmetric adjustments) and naive (unweighted, symmetric adjustments) CP (Fig. 4B), symmetrically (red) and asymmetrically (blue) adjusted intervals and where they overlap (purple) (Fig. 4C), and bias versus lengths for symmetric adjustments (red), lengths for asymmetric adjustments (blue), and upper bound lengths for symmetric adjustments from Thm. 2 (purple).

We observe that symmetric and asymmetric adjustments produce valid coverage while naive approaches do not, confirming prior work (Angelopoulos & Bates, 2021; Barber et al., 2023) (Fig. 4B). We observe that asymmetric adjustments are independent of bias (Fig. 4D) yet still produce valid prediction intervals. We observe that symmetrically adjusted interval lengths increase linearly with increasing bias, bounded by Thm. 2 (Fig. 4D). Our results suggest that even if predictions drift "far away" from the ground truth values, asymmetrically adjusted intervals are still able to maintain the same tightness and validity of intervals as if the drift had never happened.

## 5 DISCUSSION AND CONCLUSION

We will never collect perfect data in practice, or build perfect predictive models that are robust over time. Therefore, it is integral to account for these imperfections when designing practical systems. In this work, we argue that the effects of bias on CP prediction interval lengths can be mitigated by computing asymmetric adjustments as opposed to the conventional symmetric adjustments. We prove the following for $L_1$ and CQR non-conformity scores. In Thm. 2 we showed that the upper bound of the prediction interval lengths with symmetric adjustments increases by $2|b|$. In Thm. 3, we

showed that prediction interval lengths with asymmetric adjustments are not affected by bias. In Cor. 3.1, we showed the conditions when prediction interval lengths with asymmetric adjustments are guaranteed to be smaller than those of symmetric intervals. We proposed an algorithm to empirically determine the bias and showed empirical evidence using synthetic and real-life data. Our results have have important implications on accounting for algorithmic bias, while also suggesting further areas of investigation:

**Stability of bias estimation for low $n$.** Our work suggests that estimating bias based on symmetrically adjusted intervals is a straightforward, practical, and computationally efficient way to account for systematic errors in predictions. However, it is crucial to consider the impact of sample size on the reliability of these estimates when the calibration set is small (Tversky & Kahneman, 1971). Small calibration sets can lead to noisy estimates of bias (Springate, 2012). The challenges associated with limited data include under or overestimating the true bias, greater bias estimation variability, and being more susceptible to skewed, outliers, and random fluctuations. We recommend considering the uncertainty in bias estimates when applying corrections and increasing sample size to improve the reliability of bias estimates where possible.

**More complex scores.** Our analysis reveals that while simple non-conformity scores, as presented in Eq. 2 and 4, are tractable for theoretical guarantees, more complex non-conformity scores such as locally adaptive scores Papadopoulos et al. (2008; 2011); Lei et al. (2018) and CQR variants Sesia & Candès (2020) present challenges. These challenges arise due to modeling bias as a globally applied additive constant. This simplification, although useful for theoretical and empirical analysis, may overlook that biases could exhibit more intricate patterns, possibly varying across the input space or depending on specific features. For example, in time series data, we use a uniform weighting scheme that effectively reduces to split CP over each time window. Our results suggest that incorporating these techniques in more complex settings could reveal interesting behaviors and could help design more bias-robust scores.

**Covariate shift and Bias.** Our work suggests correcting for bias using a globally applied constant to the predictions can significantly reduce the interval lengths. However, our approach does not account for situations where the predicted distribution changes between calibration and test datasets. Prior work on CP under covariate shift (Tibshirani et al., 2019) weighted predictions by a probability proportional to their likelihood ratio. However, when calibration predictions are "far away" from the expected test predictions, the likelihood ratio may be very small or zero. Thus, it is impossible to perform a covariate shift without significant overlap between the calibration and expected test predictions when the expected bias is large. Our work suggests exploring both bias correction and covariate shift together could lead to tighter and more reliable prediction intervals for these situations.

ACKNOWLEDGEMENTS

[redacted]

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

# A PROOFS

## A.1 PROOF OF THEOREM 2

Using Eq. 2, we show the behavior of prediction interval lengths with symmetric adjustments when predictions are biased $\hat{Y}_i^b = \hat{Y}_i^0 + b$ where $b \in \mathbb{R}$. We show the behavior of symmetric adjustments under 1) no bias, 2) large negative bias, 3) large positive bias, 4) small negative bias and 5) small positive bias, and compare the resulting interval lengths. We leverage a property of quantiles $Q_\alpha(\hat{Y} + b) = Q_\alpha(\hat{Y}) + b$ where $b$ is a scalar value.

When $b = 0$, we can write the adjustment (Eq. 9), the prediction interval (Eq. 10), and the prediction interval length (Eq. 11).

$$s_i^0 = \max(f_{lo}(\hat{Y}_i^0) - Y_i, Y_i - f_{hi}(\hat{Y}_i^0)) \tag{8}$$

$$q^0 = Q_{1-\hat{\alpha}}(\{s_i^0\}_{i=1}^n) \tag{9}$$

$$C_{sym}(\hat{Y}_{n+1}^0) = [f_{lo}(\hat{Y}_{n+1}^0) - q^0, f_{hi}(\hat{Y}_{n+1}^0) + q^0] \tag{10}$$

$$L_{sym}(\hat{Y}_{n+1}^0) = f_{hi}(\hat{Y}_{n+1}^0) - f_{lo}(\hat{Y}_{n+1}^0) + 2q^0 \tag{11}$$

For biased predictions, the $0$ is replaced with a $b$.

Next, we examine when predictions are highly biased in the negative direction $\hat{Y}_i^{b^{--}} = \hat{Y}_i^0 + b^{--}$ where $b^{--} < Y_i - f_{hi}(\hat{Y}_i^{b^{--}}) < 0$, so $Y_i > f_{hi}(\hat{Y}_i^{b^{--}}) \geq f_{lo}(\hat{Y}_i^{b^{--}})$. The non-conformity score reduces to the $Y_i - f_{hi}(\hat{Y}_i^{b^{--}})$ because $f_{lo}(\hat{Y}_i^{b^{--}}) - Y_i < 0$ and can be written as:

$$s_i^{b^{--}} = Y_i - f_{hi}(\hat{Y}_i^{b^{--}}) = Y_i - f_{hi}(\hat{Y}_i^0) - b^{--} \leq s_i^0 - b^{--} \tag{12}$$

The adjustment can be written as:

$$q^{b^{--}} = Q_{1-\hat{\alpha}}(\{s_i^{b^{--}}\}_{i=1}^n) \leq Q_{1-\hat{\alpha}}(\{s_i^0\}_{i=1}^n) - b^{--} = q^0 - b^{--} \tag{13}$$

The length can be written as:

$$L(\hat{Y}_{n+1}^{b^{--}}) = f_{hi}(\hat{Y}_{n+1}^{b^{--}}) - f_{lo}(\hat{Y}_{n+1}^{b^{--}}) + 2q^{b^{--}} \tag{14}$$

$$\leq f_{hi}(\hat{Y}_{n+1}^0) - f_{lo}(\hat{Y}_{n+1}^0) + 2q^0 - 2b^{--} \tag{15}$$

$$= L(\hat{Y}_{n+1}^0) - 2b^{--} \tag{16}$$

The inequalities in Eq. 12, 13 and 15 hold true because $\max(f_{lo}(\hat{Y}_i^{b^{--}}) - Y_i, Y_i - f_{hi}(\hat{Y}_i^{b^{--}})) \geq Y_i - f_{hi}(\hat{Y}_i^{b^{--}})$.

Next, we examine when predictions are highly biased in the positive direction $\hat{Y}_i^{b^{++}} = \hat{Y}_i^0 + b^{++}$ where $b^{++} > Y_i - f_{lo}(\hat{Y}_i^{b^{++}}) > 0$, so $Y_i < f_{lo}(\hat{Y}_i^{b^{++}}) \leq f_{hi}(\hat{Y}_i^{b^{++}})$. The non-conformity score reduces to the $f_{lo}(\hat{Y}_i^{b^{++}}) - Y_i$ because $Y_i - f_{hi}(\hat{Y}_i^{b^{++}}) < 0$ and can be written as:

$$s_i^{b^{++}} = f_{lo}(\hat{Y}_i^{b^{++}}) - Y_i = f_{lo}(\hat{Y}_i^0) - Y_i + b^{++} \leq s_i^0 + b^{++} \tag{17}$$

$$q^{b^{++}} = Q_{1-\hat{\alpha}}(\{s_i^{b^{++}}\}_{i=1}^n) \leq Q_{1-\hat{\alpha}}(\{s_i^0\}_{i=1}^n) + b^{++} = q^0 + b^{++} \tag{18}$$

$$L(\hat{Y}_{n+1}^{b^{++}}) = f_{hi}(\hat{Y}_{n+1}^{b^{++}}) - f_{lo}(\hat{Y}_{n+1}^{b^{++}}) + 2q^{b^{++}} \tag{19}$$

$$\leq f_{hi}(\hat{Y}_{n+1}^0) - f_{lo}(\hat{Y}_{n+1}^0) + 2q^0 + 2b^{++} \tag{20}$$

$$= L(\hat{Y}_{n+1}^0) + 2b^{++} \tag{21}$$

The inequalities in Eq. 17, 18 and 20 hold true because $\max(f_{lo}(\hat{Y}_i^{b^{++}}) - Y_i, Y_i - f_{hi}(\hat{Y}_i^{b^{++}})) \geq f_{lo}(\hat{Y}_i^{b^{++}}) - Y_i$.

For the $L_1$ non-conformity score $f_{lo}(\hat{Y}_i^b) = f_{hi}(\hat{Y}_i^b) = \hat{Y}_i^b$, we can combine Eq. 16 and 21 to yield the desired result: $L(\hat{Y}_{n+1}^b) \leq L(\hat{Y}_{n+1}^0) + 2|b|$

For the CQR non-conformity score $f_{lo}(\hat{Y}_i^b) = Q_{\alpha_{lo}}(\hat{Y}_i^b)$ and $f_{hi}(\hat{Y}_i^b) = Q_{\alpha_{hi}}(\hat{Y}_i^b)$, we need to analyze when prediction have small negative bias and small positive bias - in other words, when the ground truth is between the lower and upper adjustment points.

When predictions have small negative bias $\hat{Y}_i^{b^-} = \hat{Y}_i^0 + b^-$ where $0 > b^- > Y_i - f_{hi}(\hat{Y}_i^{b^-})$ and $f_{lo}(\hat{Y}_i^{b^-}) < Y_i < f_{hi}(\hat{Y}_i^{b^-})$. The ground truth is closer to the upper bound than lower bound $f_{hi}(\hat{Y}_i^{b^-}) - Y_i < Y_i - f_{lo}(\hat{Y}_i^{b^-})$. Taking negative on both sides gives $Y_i - f_{hi}(\hat{Y}_i^{b^-}) > f_{lo}(\hat{Y}_i^{b^-}) - Y_i$. This is the same as large negative bias. The interval length reduces to Eq. 16.

When predictions have small positive bias $\hat{Y}_i^{b^+} = \hat{Y}_i^0 + b^+$ where $0 < b^+ < Y_i - f_{lo}(\hat{Y}_i^{b^+})$ and $f_{lo}(\hat{Y}_i^{b^+}) < Y_i < f_{hi}(\hat{Y}_i^{b^+})$. The ground truth is closer to the lower bound than upper bound $f_{hi}(\hat{Y}_i^{b^+}) - Y_i > Y_i - f_{lo}(\hat{Y}_i^{b^+})$. Taking negative on both sides gives $Y_i - f_{hi}(\hat{Y}_i^{b^+}) < f_{lo}(\hat{Y}_i^{b^+}) - Y_i$. This is the same as large positive bias. Thus, the interval length reduces to Eq. 21.

Combining inequalities gives the desired result: $L(\hat{Y}_{n+1}^b) \leq L(\hat{Y}_{n+1}^0) + 2|b|$

## A.2 PROOF OF THEOREM 3

We analyze the behavior of asymmetric adjustments under bias. We model the biased predictions as $\hat{Y}_i^b = \hat{Y}_i^0 + b$ where $b$ is a global constant (can be both positive and negative) added to unbiased predictions $\hat{Y}_i^0$.

First, the lower and upper scores can be written as:

$$s_{i,lo}^b = f_{lo}(\hat{Y}_i^b) - Y_i = f_{lo}(\hat{Y}_i^0) - Y_i + b = s_{i,lo}^0 + b \tag{22}$$

$$s_{i,hi}^b = Y_i - f_{hi}(\hat{Y}_i^b) = Y_i - f_{hi}(\hat{Y}_i^0) - b = s_{i,hi}^0 - b \tag{23}$$

Next, the lower and upper adjustments can be written as:

$$q_{lo}^b = Q_{1-\hat{\alpha}_{lo}}(\{s_{i,lo}^b\}_{i=1}^n) = Q_{1-\hat{\alpha}_{lo}}(\{s_{i,lo}^0\}_{i=1}^n) + b = q_{lo}^0 + b \tag{24}$$

$$q_{hi}^b = Q_{1-\hat{\alpha}_{hi}}(\{s_{i,hi}^b\}_{i=1}^n) = Q_{1-\hat{\alpha}_{hi}}(\{s_{i,hi}\}_{i=1}^n) - b = q_{hi}^0 - b \tag{25}$$

Finally, the lengths when predictions are biased can be simplified:

$$L_{asym}(\hat{Y}_{n+1}^b) = f_{hi}(\hat{Y}_{n+1}^b) - f_{lo}(\hat{Y}_{n+1}^b) + q_{hi}^b + q_{lo}^b \tag{26}$$

$$= f_{hi}(\hat{Y}_{n+1}^0) + b - f_{lo}(\hat{Y}_{n+1}^0) - b + q_{hi}^0 + b + q_{lo}^0 - b \tag{27}$$

$$= f_{hi}(\hat{Y}_{n+1}^0) - f_{lo}(\hat{Y}_{n+1}^0) + q_{hi}^0 + q_{lo}^0 \tag{28}$$

$$= L_{asym}(\hat{Y}_{n+1}^0) \tag{29}$$

The results in Eq. 26 indicate that asymmetric adjustments are not affected by bias $b$.

### A.2.1 PROOF OF ALG. 1 CONVERGENCE

We seek to minimize the objective function:

$$f(b_{eff}) = \max(\{L_{sym}(\hat{Y}_i^b - b_{eff})\}_{i=1}^n) \tag{30}$$

$$= \max(\{L_{sym}(\hat{Y}_i^0 + b - b_{eff})\}_{i=1}^n) \tag{31}$$

First, we can derive the symmetrically adjusted interval lengths with bias $b \in \mathbb{R}$ based on the techniques and results from App. A.1. Initially, the predictions are assumed to be biased. The non-conformity scores in canonical form (Eq. 2) when predictions are positively and negatively biased with $b^+ > 0$ and $b^- < 0$ are given by:

$$s_i^{b^+} = f_{lo}(\hat{Y}_i^{b^+}) - Y_i = f_{lo}(\hat{Y}_i^0) - Y_i + b^+ = s_i^0 + b^+ \tag{32}$$

$$s_i^{b^-} = Y_i - f_{hi}(\hat{Y}_i^{b^-}) = Y_i - f_{hi}(\hat{Y}_i^0) - b^- = s_i^0 - b^- \tag{33}$$

where $s_i^0$ is the non-conformity score without bias. The inequality is replaced with an equality because predictions are assumed to be biased during the optimization process. Specifically, $\max(f_{lo}(\hat{Y}_i^{b^+}) - Y_i, Y_i - f_{hi}(\hat{Y}_i^{b^+})) = f_{lo}(\hat{Y}_i^{b^+}) - Y_i$ under positive bias and $\max(f_{lo}(\hat{Y}_i^{b^-}) - Y_i, Y_i - f_{hi}(\hat{Y}_i^{b^-})) = Y_i - f_{hi}(\hat{Y}_i^{b^-})$ under negative bias. Next, the adjustments can be written as:

$$q^{b^+} = Q_{1-\hat{\alpha}}(\{s_i^{b^+}\}_{i=1}^n) = Q_{1-\hat{\alpha}}(\{s_i^0\}_{i=1}^n) + b^+ = q^0 + b^+ \tag{34}$$

$$q^{b^-} = Q_{1-\hat{\alpha}}(\{s_i^{b^-}\}_{i=1}^n) = Q_{1-\hat{\alpha}}(\{s_i^0\}_{i=1}^n) - b^- = q^0 - b^- \tag{35}$$

where $q^0 \in \mathbb{R}$ is the symmetric adjustment for predictions with no bias. Combining the two equations gives a more general form of adjustments for positive and negative bias:

$$q^b = q^0 + |b| \tag{36}$$

Thus, symmetrically adjusted interval lengths with bias can be written as:

$$L_{sym}(\hat{Y}_{n+1}^b) = f_{hi}(\hat{Y}_{n+1}^b) - f_{lo}(\hat{Y}_{n+1}^b) + 2q^b \tag{37}$$

$$= f_{hi}(\hat{Y}_{n+1}^b) - f_{lo}(\hat{Y}_{n+1}^b) + 2q^0 + 2|b| \tag{38}$$

Using Eq. 38, we can recast the objective function as follows:

$$f(b_{eff}) = \max(\{L_{sym}(\hat{Y}_i + b_{eff})\}_{i=1}^n) \tag{39}$$

$$= \max(\{L_{sym}(\hat{Y}_i^0 + b - b_{eff})\}_{i=1}^n) \tag{40}$$

$$= \max(\{f_{hi}(\hat{Y}_i^0) - f_{lo}(\hat{Y}_i^0) + 2q^0\}_{i=1}^n) + 2|b - b_{eff}| \tag{41}$$

In Eq. 41, the terms inside the $\max$ function and $b$ are data-dependent constants and are not dependent on $b_{eff}$. Thus, minimizing Eq. 39 results in minimizing a translated (horizontally and vertically) absolute value function (Eq. 41). This problem is convex, not differentiable at $b_{eff} = b$, and has a global minimum at $b$. When $b_{eff} \neq b$, the sub-gradient of $f(b_{eff})$ is $-2$ for $b_{eff} < b$ and $2$ for $b_{eff} > b$. A standard convergence proof follows.

Let $b_{eff,k}$ be the $k$-th iteration of gradient descent. The update rule is: $b_{eff,k+1} = b_{eff,k} - \gamma \nabla f(b_{eff,k})$. For any step size $\gamma > 0$, we have:

1. If $b_{eff,k} > b$, $b_{eff,k+1} = b_{eff,k} - 2\gamma$, moving towards $b$
2. If $b_{eff,k} < b$, $b_{eff,k+1} = b_{eff,k} + 2\gamma$, moving towards $b$

Thus, the distance to the optimum decreases in each iteration:

$$|b_{eff,k+1} - b| \leq |b_{eff,k} - b| - 2\gamma \tag{42}$$

After $k$ iterations, the distance to the optimum is at most:

$$|b_{eff,k} - b| \leq |b_{eff,0} - b| - 2\gamma k \tag{43}$$

Setting this to $\epsilon$ and solving for $k$ yields

$$k \geq \frac{|x_0 - b| - \epsilon}{2\gamma} \tag{44}$$

Thus gradient descent converges to the global minimum $b_{eff} = b$ with rate $O(1/k)$

## B  DATA DISTRIBUTIONS

We show the data distributions for experiments in Sec. 4 in Fig. 5 and 6.

## C  SPARSE CT FOR RADIOTHERAPY PLANNING DETAILS

We use a de-identified CT dataset of 20 patients retrospectively treated with radiotherapy at [redacted]. This research was conducted using an approved institutional review board protocol. For each patient, we generate 10 digitally reconstructed radiographs (DRR) from the ground truth CT scan using the TIGRE toolbox Biguri et al. (2016). The DRRs simulate image acquisition from a cone-beam geometry. We simulate physical randomness (beam angle variability and sensor noise) by generating DRRs with 3% noise and 50 random projections between 0 and 360 degrees. We use a self-supervised model, Neural Attenuation Fields (NAF), for reconstruction (Zha et al., 2022). We use the Radiation Planning Assistant (RPA, FDA 510(k) cleared), a web-based tool for radiotherapy planning. (Aggarwal et al., 2023; Court et al., 2023; Kisling et al., 2018). RPA automates treatment planning on CT images and provides dose and plan reports for clinics in low-and-middle-income countries (Aggarwal et al., 2023; Court et al., 2023; Kisling et al., 2018). The number of projections was increased from 2 to 50 until organ boundaries in the reconstructed volumes were perceptually discernible in the reconstruction by the RPA. We use the default parameter setting in NAF (Zha et al., 2022) and introduce computational randomness through random initializations of NAF (Sünderhauf et al., 2023; Lakshminarayanan et al., 2017).

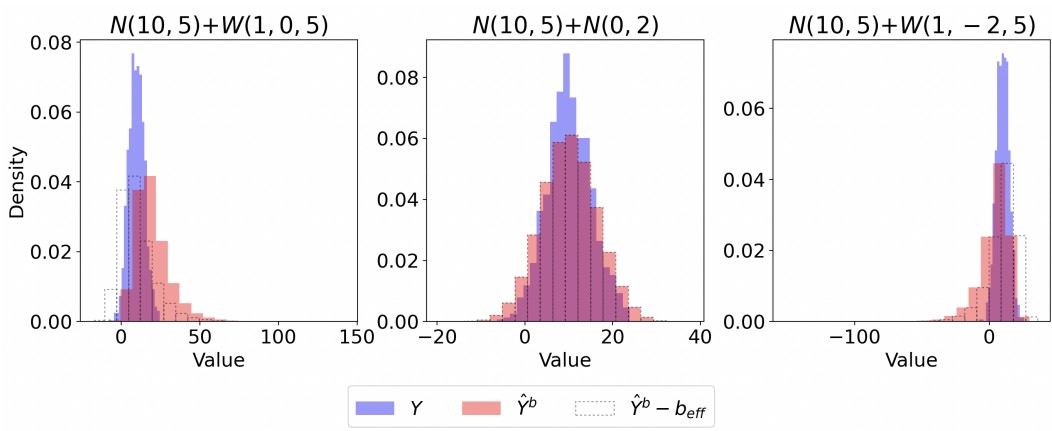

Figure 5: **Data distribution for synthetic experiments with skewed and noisy predictions.** We use $N(10, 5)$ to simulate the ground truth (blue) distribution and added $N(0, 2)$, $W(1, 0, 5)$, and $-W(1, -2, 5)$ to the ground truth to simulate a un-, left- and right- skewed predictions (red). The parameter descriptions can be found in the scipy.stats documentation. The dotted histograms indicate the "unbiased" predictions $\hat{Y}^b - b_{eff}$ where $b_{eff}$ is the empirical effective bias estimated through a simple optimization procedure.

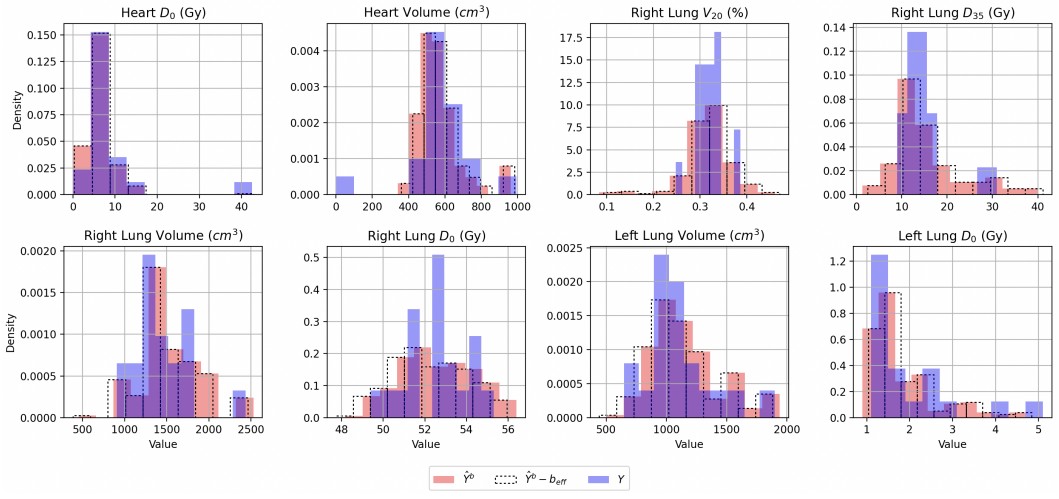

Figure 6: **Data distribution for sparse view computed tomography (sparse CT) reconstruction applied to downstream radiotherapy planning.** Including max dose to the heart (Heart $D_0$), heart volume, volume of right lung receiving 20Gy of dose (right lung $V_{20}$), dose to 35% relative volume of the right lung (right lung $D_{35}$), right lung volume, left lung volume, max dose to left lung (Left Lung $D_0$), and volume of the body. The predictions $\hat{Y}^b$ and ground truth $Y$ are shown in red and blue. The dotted histograms indicate the "unbiased" predictions $\hat{Y}^b - b_{eff}$ where $b_{eff}$ is the empirical effective bias estimated through a simple optimization procedure in Alg. 1.