# OpenReview forum: "Regression Conformal Prediction under Bias"
_ICLR.cc/2025/Conference — ICLR 2025 Conference Withdrawn Submission_

### Official Review · Reviewer_V4yg · 2024-10-23

**Soundness:** 2
**Presentation:** 2
**Contribution:** 1
**Rating:** 1
**Confidence:** 3

**Summary:**

This paper investigates how bias, defined as the systematic deviation from ground truth, affects Conformal Prediction (CP) intervals in regression tasks. CP is used for uncertainty quantification in machine learning models. The authors focus on two adjustment methods for CP intervals: symmetric (where both sides of the interval are adjusted equally) and asymmetric (where adjustments can be unequal). Through theoretical and empirical analyses, they show that:
1. Symmetrically adjusted intervals increase in length by 2|b|, where b is the bias.
2. Asymmetrically adjusted intervals are unaffected by bias.
3. Under certain conditions, asymmetric intervals are tighter than symmetric ones.
Their findings suggest that asymmetric intervals maintain their "tightness" even under biased predictions, unlike symmetric intervals that inflate in length. These conclusions are validated with real-world tasks in CT reconstruction and weather forecasting, highlighting the potential for more bias-robust machine learning systems.

**Strengths:**

Practical Consideration of CP Efficiency: The authors address the critical issue of efficiency in conformal prediction (CP) sets, specifically focusing on the size of prediction intervals, which is relevant in real-world applications. By exploring the impact of bias on CP intervals, the authors attempt to improve the accuracy and reliability of predictions, which is essential for high-stakes tasks.

Experiments: The paper demonstrates multiple real-world experiments, such as sparse-view CT reconstruction and weather forecasting.

**Weaknesses:**

Inconsistent Notations and presentation: The paper suffers from unclear and inconsistent notations. For instance, in equation (1), it's not explicit which bias is being addressed --- it is not clear how the randomness in the ground truth is being dealt with.  Additionally, \hat{Y_i^b} is ambiguously defined; it is not clear whether these quantities are scalars or they are allowed to be a set of values. For instance, in the first paragraph of Section 3 the Y_i is treated as scalars and in the second paragraph, they use Y_i = {Y_{ij}} as a vector. Besides, the score definition proposed by Romano et al.[1] is presented inaccurately, which raises concerns about the foundation of the theoretical analyses. The overall readability of the paper is not good.

Limited Insight from Corollary: While Theorems 1 and 2 seems a bit straightforward and is a direct consequence of the linear assumptions. The utility of Corollary 3.1 is not clear, the RHS of eq. (6) is independent on b, and the LHS is dependent on b? It is not clear how the authors are setting L_{asymb} \leq L_{sym} in the proof.

Missing Theoretical Justification for the Algorithm: The authors do not provide a theorem guaranteeing that the proposed algorithm retains the validity of the coverage guarantee for CP intervals. This omission is significant as it leaves a gap in understanding whether the method meets one of CP's fundamental requirements.

Lack of Coverage Probability in Simulations: The simulations fail to report the coverage probability, which is a critical metric for evaluating the reliability of proposed algorithm. This weakens the experimental validation of the proposed methods.

Comparison with related work: There are some related work which also aims to reduce the length of the conformal intervals [2,3]. It would be nice to compare the contribution of this work with these two related works.

[1.] Romano, Yaniv, Evan Patterson, and Emmanuel Candes. "Conformalized quantile regression." Advances in neural information processing systems 32 (2019).

[2.] Xie, Ran, Rina Foygel Barber, and Emmanuel J. Candès. "Boosted Conformal Prediction Intervals." arXiv preprint arXiv:2406.07449 (2024).

[3.] Liang, Ruiting, Wanrong Zhu, and Rina Foygel Barber. "Conformal prediction after efficiency-oriented model selection." arXiv preprint arXiv:2408.07066 (2024).

**Questions:**

Please have look at the weakness section.

---

### Official Review · Reviewer_LHC9 · 2024-10-24

**Soundness:** 2
**Presentation:** 3
**Contribution:** 1
**Rating:** 1
**Confidence:** 5

**Summary:**

The paper studies how a systematic homoscedastic bias is present in the point prediction. It studies how it influences the efficiency of prediction intervals produced by inductive conformal prediction (ICP) and conformalized quantile regression (CQR), explicitly comparing the effect of symmetric and asymmetric adjustments. For ICP, the impact of using the absolute residual (symmetric adjustment) and the residual (asymmetric adjustment, controlling the coverage on the left and right) is compared. For CQR, similarly.

They show that in the case of simple homoscedastic bias, $b$: 1) the upper bound for the size of the intervals of the biased predictions, which of the size of the intervals without prejudice and $2|b|$; 2) the asymmetric adjusted prediction interval is of the same size as there was no bias, and 3) they show under which condition the asymmetric or symmetric adjustment will result in smaller prediction intervals (however, this requires knowledge of the bias).

**Strengths:**

- **Clear and Comprehensive:** The CP mechanisms, the derivation of the bias effects, and the comparison between symmetric and asymmetric intervals are clearly explained.
- **Formalizing Good Message:** Recently, in the CP field/community, asymmetric adjustments or conformal prediction intervals with symmetric (the same) density are used beneath the lower bound and above the upper bound, which is appropriate for the end-user from an interpretation standpoint. Additionally, there is occasional observation of performance gains using asymmetric adjustment, as the paper rightfully mentions. Formalizing and theorizing these observations is undoubtedly beneficial for the field and practitioners.

**Weaknesses:**

- **Elementary Theoretical Findings**: While the theoretical contributions are sound, the results may be considered elementary. The core findings, such as the impact of bias on symmetric intervals and the lack of bias influence on asymmetric intervals, are straightforward and follow from the basic principles of CP. Though helpful, these insights do not introduce particularly advanced or novel theoretical complexity, which may limit the paper's appeal to a more specialized audience seeking more profound theoretical innovations.
- **Simplified Bias Assumptions**: The assumption of bias as a constant noise term following the same global distribution across all predictions may not adequately capture more complex forms of bias present in real-world data, such as feature-specific or covariate-dependent biases. The paper also mentions on lines 089-094 different reasons for these biases. However, they result in way more complex biases.
- **Limited Discussion on More Complex Non-Conformity Scores**: While the paper acknowledges that more complex non-conformity scores may require different approaches, it does not explore these in-depth, potentially limiting the generality of the findings for more advanced CP methods.
- **Limited Synthetic Experiment Settings:** The paper only evaluates experiments where $n$ is large, and the data is generated from symmetrical distributions. However, this is problematic because the strong point of the symmetric adjustment is that it can better leverage asymmetries (skewness) of the aleatoric uncertainty distribution (noise distribution on the label).
- **Wrong Claims:** Your statement on line 266 is incorrect. When the number of samples is large and no bias is present, the length of prediction intervals generated by a symmetric adjustment is approximately equal to the ones generated from asymmetric adjustments. See the above point.
- **Section 3.1 is too bloated:** Given your bias, you could just take the mean of the calibration set's error, to retrieve it.

**Questions:**

- How do the paper results get used in practice? Since any learning algorithm perfectly mitigates the bias that you present. So, in your case, the bias is not present in the training set but is in the calibration set (so that it can account for this). Can you provide a use case where this occurs?
- You need to correct the table reference; you see Tab. 4.2.1, while in the caption, it is Table 1. The same is true for the references to the figures, where you use Fig. X, and the caption states Figure X.  Change either the captions or the references.

---

### Official Review · Reviewer_Wuts · 2024-10-28

**Soundness:** 3
**Presentation:** 3
**Contribution:** 2
**Rating:** 5
**Confidence:** 3

**Summary:**

The proposed method aims to improve uncertainty quantification in the presence of model bias. The authors show that non-symmetric prediction intervals may be more robust to model bias than their symmetric counterparts. The claims are supported by theoretical and empirical results.

**Strengths:**

- As CP applies to any given model, the results may help establish a good trade-off between model flexibility and efficiency in applications where data are scarce.
- The algorithm for estimating the bias from the obtained interval is interesting.

**Weaknesses:**

- Non-symmetric CP is not new. The paper's contribution would be better introduced by adding an intuitive explanation of why *asymmetric adjustments have been theorized to yield longer interval lengths as a consequence of stronger guarantees (Romano et al.,2019)* but *has also been empirically observed that asymmetric adjustments yield tighter intervals than symmetric ones.*
- The authors focus on the setup where comparable bias is present in the calibration and test samples. They should comment on whether it would be more efficient to 1) improve the underlying point-prediction model or 2) consider more flexible conformity scores (e.g. adding and training a free shifting term to the conformity scores).
- Asymmetric intervals can be defined by splitting the calibration samples according to the residual signs and then evaluating two sample quantiles. It would be helpful to see a theoretical comparison of the gap between the proposed method and such a naive approach as a function of the size of the calibration set.

**Questions:**

- Is Romano et al. 2029 the only work where non-symmetric intervals are used and tested?
- What happens if an adaptive scheme is used to compute the interval? E.g. if the symmetric constant intervals are replaced by reweighted intervals as in Papadopoulous 2008? In particular, how would Theorem 2 change?
- Would it be possible to compute an upper bound of the bias using the gap between symmetric and asymmetric intervals?
- Can the proposed approach help study the non-exchangeable situation where the bias is only present at test time? In other words, can one obtain an approximated version of Lemma 1 and Theorem 2?
- Have you run any experiments on Bias estimation?

---

### Official Review · Reviewer_fjtq · 2024-11-03

**Soundness:** 2
**Presentation:** 2
**Contribution:** 2
**Rating:** 3
**Confidence:** 4

**Summary:**

This paper studies the efficiency of conformal prediction when the predicted value has a systematic bias, aiming to understand the effect of symmetric/asymmetric quantile adjustment on the corresponding
conformal prediction length. Under a stylized model, theories are
developed to understand the behavior of symmetric/asymmetric quantile adjustment.
The theory is then evaluated on synthetic and real data.

**Strengths:**

The paper studies an important question and makes an interesting
observation. Some theoretical attempts have also been made to understand the
observation.

**Weaknesses:**

The paper's main theories (Theorem 2 and 3) are under highly stylized models and the statement lacks rigor (please correct me
if I misunderstood anything). For example, Theorem 2 is stated for prediction intervals in the generic form of Eq. (2), but it appears that the proof only focuses on the case of residual errors and CQR. For another example, the proof of Theorem 2 seems to use the fact that
$f_{\text{hi}}(\hat Y_i^{b--}) = f_{\text{hi}}(\hat Y_i^{0})+b^{--}$, which is not stated as an assumption. The theorem also appears to assume that the bias is sufficiently large in magnitude.


I am also not following the form of CQR used in this paper; in my understanding,
CQR fits conditional quantiles of $Y$ given $X$ by minimizing the pin-ball loss,
and the resulting fitted conditional quantile  function operates $X_i$ instead of the point prediction of $\hat Y_i$
(unless the quantile is obtained in a specific form centered around a point prediction).
I am also confused by the description of CQR in lines 194-196 --- what is $n_s$, what is the set of samples $\hat Y^b_i$
and how are they generated. If the prediction is deterministic, does this reduce to the residual error case? Please let me know if I misunderstood anything.

**Questions:**

My main questions are in the weakness section. Below is a question regarding the proof: In the proof of Theorem 2 (line 712), I am confused since it is first stated that $Y_i - f_{\text{hi}}(\hat Y_i^{b^{--}}) <0$ and then
$Y_i > f_{\text{hi}}(\hat Y_i^{b^{--}})$.

---

### Note · Authors · 2024-11-15

**Comment:**

After considering the comments from reviewers, we have decided to withdraw our paper. We thank the reviewers for their insightful comments and suggestions.

**Withdrawal Confirmation:**

I have read and agree with the venue's withdrawal policy on behalf of myself and my co-authors.